# Factors Associated with Colostrum Quality, the Failure of Transfer of Passive Immunity, and the Impact on Calf Health in the First Three Weeks of Life

**DOI:** 10.3390/ani13111740

**Published:** 2023-05-24

**Authors:** Katharina Lichtmannsperger, Christina Hartsleben, Magdalena Spöcker, Nicole Hechenberger, Alexander Tichy, Thomas Wittek

**Affiliations:** 1Department for Farm Animals and Veterinary Public Health, University Clinic for Ruminants, University of Veterinary Medicine Vienna, Veterinaerplatz 1, 1210 Vienna, Austria; christina.hartsleben@yahoo.de (C.H.); m.spoecker@gmx.at (M.S.); thomas.wittek@vetmeduni.ac.at (T.W.); 2Animal Health Service Salzburg, Bundesstraße 6, 5071 Wals-Siezenheim, Austria; nicole.hechenberger@salzburg.gv.at; 3Department for Biomedical Sciences, Bioinformatics and Biostatistics Platform, University of Veterinary Medicine Vienna, Veterinaerplatz 1, 1210 Vienna, Austria; alexander.tichy@vetmeduni.ac.at

**Keywords:** colostrum quality, poor-quality colostrum, bacterial contamination, FTPI, morbidity, odds ratio, calf diarrhea, navel illness, bovine respiratory disease, health event

## Abstract

**Simple Summary:**

Calves rely on passive immunization with significant quantities of high-quality colostrum within the first few hours of life. If immunoglobulin transfer after birth fails, calves face the failure of transfer of passive immunity. FTPI has been known to lead to high morbidity and mortality rates. Therefore, FTPI constitutes a major animal welfare issue. The objectives of this study were to evaluate the factors associated with colostrum quality and FTPI in calves from dairy farms in Austria and to assess the associations between disease occurrence and FTPI. The number of lactations of the dam and the time lag between parturition and colostrum harvest were significantly associated with low colostrum quality. Colostrum quantity and colostrum quality were factors significantly associated with FTPI. Calf morbidity rates, especially for diarrhea, were significantly associated with FTPI. The present investigation underlines the importance of improving farmers’ awareness of colostrum management, especially of the most substantial factors, including colostrum quality and colostrum quantity; it further elucidates the consequences of FTPI for disease occurrence.

**Abstract:**

The objectives of this study were to evaluate factors associated with colostrum quality and FTPI in calves from dairy farms in Austria and to assess the associations between disease occurrence and FTPI in calves. In total, 250 calves and their colostrum samples originating from 11 dairy farms were included in the study. All calves born between September 2021 and September 2022 were included. Blood samples were collected between the third and the sixth day of age. The farmers were trained in disease detection and recorded any health events within the first three weeks of age daily. Multiparous cows (>3 lactation) and colostrum harvesting within the first 2 hours after parturition were significantly associated with good colostrum quality (>22% Brix). Colostrum quantity (≥2 L) and quality (≥22% Brix) acted as protective factors against FTPI (serum Brix ≥ 8.4%) with odds ratios of OR = 0.41 and OR = 0.26, respectively. Calves facing any health event (diarrhea, navel illness, bovine respiratory disease, abnormal behavior) in the first three weeks of life had a higher probability of FTPI. Calves exhibiting diarrhea in the first 3 weeks of life were associated with having FTPI (OR = 2.69). The results confirm the current recommendations for good colostrum management practices and the impact of FTPI on calf morbidity.

## 1. Introduction

Calves rely on passive immunity based on immunoglobulins (IgGs) originating from the colostrum, since the cotyledonary synepitheliochorial placenta type of cows acts as a barrier for immunoglobulin transfer during pregnancy [1]. Feeding with sufficient quantities of high-quality colostrum immediately after birth is the most important protective factor in preventing calves from suffering from the failure of transfer of passive immunity (FTPI) [1]. Additionally, the colostrum contains many other essential constituents, for instance, growth factors, hormones, antimicrobial factors, leucocytes, oligosaccharides, mRNA, and nutrients; the actual role of all these constituents is not yet clearly understood [1]. High-quality colostrum is defined as having high immunoglobulin concentrations (≥50 g/L) and low bacterial contamination (total plate counts < 100,000 cfu/mL; coliform counts < 10,000 cfu/mL) [1,2,3]. Feeding with low-quality colostrum (<50 g/L) is known to be one of the factors significantly associated with FTPI, with an odds ratio of 10.7 (OR = 10.7, 95% confidence interval CI = 4.7–24.2) [4]. The apparent efficiency of immunoglobulin absorption tends to be higher in colostrum with lower levels of bacterial contamination [5]. Additionally, calves fed a second colostrum meal were less likely to suffer from FTPI. Of the 4336 investigated calves, 9.4% of those that received two meals of colostrum experienced FTPI and 22.2% of the calves that received one meal of colostrum experienced FTPI [6]. The definition of FTPI is a serum IgG level below 10 mg/mL, which is equal to serum Brix levels of 8.4% [7,8,9]. In a meta-analysis of the consequences of FTPI, the adjusted risks (95% CI) for mortality, bovine respiratory disease (BRD), diarrhea, and overall morbidity in the case of FTPI were 2.12 (1.43–3.13), 1.75 (1.50–2.03), 1.51 (1.05–2.17) and 1.91 (1.63–2.24), respectively [10]. A study from Germany noted that calf mortality rates (definition of calf mortality: 2 days to 6 months of age) of above 5% were reported on farms with more than 25% of the calves suffering from FTPI [11]. A study from New Zealand including pasture-based dairy systems also detected greater odds of farmer-recorded animal health events (OR = 1.68) in calves with FTPI [12]. Besides the fact that FTPI leads to significant economic losses due to high morbidity and high mortality, it constitutes a severe animal welfare issue [13]. Studies from different countries show that FTPI occurs frequently, and the number of calves affected with FTPI differs substantially between studies. For instance, 14.1% of the included calves were affected in a study from Scotland (N included farms = 38, N included calves = 370), 27.0% in Germany (2 farms; 216 calves), 33.0% in New Zealand (107 farms; 3819 calves), 43.5% in Switzerland (141 farms; 373 calves), 41.9% in Australia (23 farms; 253 calves), and 19.2% in the USA (413 farms; 2030 calves) [4,14,15,16,17,18,19]. Low colostrum quality is one of the major risk factors associated with FTPI; it is influenced by multiple factors, which can be partially influenced by the farmer through management. Such management-related factors include the time lag between parturition and colostrum harvest, the presence of the dam during colostrum harvest, colostrum storage procedure, and heat treatment of the colostrum [20,21,22,23]. Cow-related factors, such as the number of lactations, genetic parameters, dry period length, ante-partum milk leakage, colostrum quantity, the metabolic status of the cow, and udder health have been noted to influence colostrum quality [4,24,25,26,27,28]. Environmental factors, such as the season of calving and the temperature–humidity index, have been proven to have an impact. There are contradictory results regarding immunoglobulin concentrations, whereas other factors (e.g., management-related and cow-related factors) seem to have stronger effects [26,29].

The objectives of this study were to evaluate the factors associated with colostrum quality and serum Brix values in calves from dairy farms in Austria and to assess whether calf diarrhea, navel illness, and BRD are associated with FTPI within the first three weeks of life.

We hypothesized that (1) multiple factors are associated with a low colostrum quality and (2) a low serum Brix level and (3) that calves experiencing diarrhea, navel illness, and/or BRD have a higher probability of exhibiting FTPI.

## 2. Materials and Methods

### 2.1. Ethical Considerations

This study was approved by the Ethics and Animal Welfare Committee (ETK) of the University of Veterinary Medicine, Vienna, and the Austrian national authorities, according to § 26 of the Tierversuchsgesetz 2012—TVG 2012 (GZ.: 2021-0.644.875).

### 2.2. Farm and Animal Selection

In total, 11 farms in the Austrian federal district of Salzburg volunteered to take part in the study; 250 calves born between 17 September 2021 and 30 September 2022 and their corresponding colostrum samples were included in the study. None of the included farms performed any kind of heat treatment procedure, such as colostrum pasteurization. Additionally, the farms did not use any kind of colostrum replacer.

### 2.3. Cows

In total, 250 cows were included; of these cows, 213 (85.2%) were healthy throughout the whole dry period and at calving and 11 cows (4.4%) showed a disease before/at parturition. Additionally, 2 (0.8%) of the 250 cows suffered from acute mastitis during the dry period, 1 cow (0.4%) showed retained fetal membranes, 7 (2.8%) suffered from clinical hypocalcemia, and 1 cow suffered from moderate lameness (0.4%). Moreover, 21 cows received a prophylactic treatment with an oral calcium bolus or a Vitamin D_3_ injection in combination with oral calcium (N = 8). The cows were grouped as suffering from disease during the dry period or at calving (“disease yes”) or not (“disease no”). In total, 15 cows received a vaccination against bovine rota-/corona virus and *E. coli* and 15 cows were vaccinated against a herd-specific salmonella strain. Of the 53 animals (10.0%) showing ante-partum milk leakage, 22 (41.5%) were primiparous cows, 17 (32.1%) received intramammary antibiotics, 4 received an internal teat sealant (ITS) (7.6%), 1 (1.9%) received no medication, and 9 (17.0%) received intramammary antibiotics and an additional ITS, respectively. Ante-partum milk leakage was defined as colostrum leakage during parturition and/or immediately before the first colostrum harvest after calving. Calving assistance was categorized according to three categories: (1) normal calving, i.e., spontaneous delivery with no assistance, (2) assisted calving, with one person assisting at the calving, (3) severe, with >1 person assisting at the calving and/or a vet being consulted.

### 2.4. Calves

All 250 calves were separated from their mothers within 1 hour of birth. The breeds were Simmental (N = 118; 59.3% female and 40.7% male), Pinzgauer (N = 65; 33.9% female and 66.1% male), and cross-breed calves (N = 44; 45.4% female and 54.6% male). All calves underwent a physical examination on the day of the blood sample collection (3rd to 6th days of age); details of the blood sample collection are summarized in Section 2.6. The time point of the blood sample collection was restricted to 3 to 6 days of age, as described elsewhere (7). The results from the physical examination included a general assessment (general behavior, posture, body condition scoring, hair, skin turgor, and mucus membranes), navel, lung, heart, and abdominal assessments (including auscultation), and a fecal assessment. The detailed assessment methods are summarized in Section 2.7.

### 2.5. Colostrum Quality Assessment

#### 2.5.1. Colostrum Brix Measurements

Following calving, the colostrum was routinely harvested (via a milking machine or hand milking) by the farmers. Subsequently, the colostrum was collected at the point of calf feeding. For example, for farmers feeding the colostrum using a feeding bucket, the sample was collected directly from the feeding bucket. One sterile 15 mL tube was filled with colostrum with the collector wearing disposable gloves. The colostrum sample was immediately frozen at −20 °C at the farm. All colostrum samples were transported frozen on ice in a polystyrene box to the diagnostic laboratory of the University Clinic for Ruminants, Vienna. In the lab, all samples were thawed in the refrigerator at 5–7 °C and vortexed for 5–10 s (VF2^®^, IKA^®^-Werke GmbH & Co. KG, Staufen,, Germany). Subsequently, the Brix refractometer (0 to 85% Brix; HM-DREF-1^®^, Hebesberger Messtechnik, Neuhofen, Austria) was calibrated using deionized water. Calibration was carried out routinely at the beginning of the analysis and following the measurement of 10 colostrum samples. After calibration, the colostrum was pipetted onto the prism using a 1-way 2 mL plastic pipette. The Brix percentage was recorded. In total, 245 of the 250 colostrum samples could be measured, and 5 samples were excluded due to technical reasons (i.e., error message on the refractometer).

#### 2.5.2. Bacterial Contamination

Bacterial contamination was assessed in the diagnostic laboratory of the University Clinic for Ruminants, Vienna. Bacterial contamination was assessed using cut-off values of 100,000 colony-forming units (cfu) per ml and 10,000 cfu/mL for total plate counts (TPC) and coliform counts, respectively (2). In brief, the colostrum samples were thawed in the refrigerator at 5–7 °C and vortexed for 5–10 s (VF2^®^, IKA^®^-Werke GmbH & Co. KG, Staufen, Germany). A 1:10 dilution series using 900 µL of sterile 0.9% physiological sodium chloride solution (B. Braun Melsungen AG, Melsungen, Germany) and 100 µL of native colostrum or the corresponding dilution was prepared in 1.5 mL Eppendorf tubes. Subsequently, 100 µL of native colostrum (1:10) and dilution 1 (1:100) and dilution 2 (1:1000) was pipetted on the Columbia agar plates (containing 5% sheep blood) to assess the TPC. Additionally, 100 µL of the native colostrum (1:10) and dilution 1 (1:100) was plated on MacConkey agar for coliform counts. All agar plates were incubated under aerobic conditions at 37 °C for 18–24 h. Photographs were taken of all agar plates and the colonies were counted using the free available Fiji Software (Fiji^®^, ImageJ). All visible colonies on the Columbia agar were counted. If the cfu levels were above 300 cfu/plate, the colostrum sample was investigated a second time after an additional dilution step (1:10,000). If the total number of colonies per plate still exceeded 300 cfu/plate; the sample was not further investigated and was considered to be non-assessable (“n.a.”). The cfu levels were multiplied with the respective dilution factor and cfu values were given as the cfu per milliliter of colostrum (cfu/mL). This study was conducted alongside another investigation by our group on colostrum management practices in Austria (Hechenberger et al., unpublished).

### 2.6. Serum and Plasma Brix Measurements

Serum and plasma samples were collected between the 3rd and the 6th day of age by jugular venipuncture using an 18-gauge needle and a Vacutainer system with serum (clot-activator) and EDTA tubes (Vacuette^®^, Greiner Bio-One International GmbH, Kremsmünster, Austria), respectively. This investigation was performed together with an evaluation of the point-of-care testing used to test for FTPI under field conditions (Hartsleben et al., under review). EDTA and serum tubes were centrifuged at 1500× *g* for 10 min at 20–25 °C at the farm of origin (CGOLDENWALL 800D Electric Centrifuge Medical Lab Centrifuge 4000 rpm with CE 6 × 20 mL, Zhengzhou Jin Chen Electronic Technology Co., Ltd., Zhengzhou, China). Subsequently, the Brix values were determined using a digital Brix refractometer (MA871 Refractometer, Hebesberger, Neuhofen, Austria) under field conditions. FTPI was defined as serum Brix values below 8.4% [7].

### 2.7. Definition of Calf Diseases

The calves underwent a complete physical examination by a veterinarian before blood sample collection (3rd to 6th days of age, see Section 2.4). A shortened physical examination was carried out by the farmers daily. Before the study, all farmers received individual training from one of the authors (C.H.) on how to perform the shortened physical examination using standard operating procedures (SOPs) and checklists. At least once per day, all calves were examined by the trained farmer. For each calf, an individual “calf card” with checklists was designed and fixed on the box of the respective calf (for details, see Appendix A). These calf cards were created by the authors based on the clinical propaedeutics and the calf cards of the Swiss calf health service (Kälbergesundheitsdienst) [30]. Any pathological findings were noted on the calf card as described on the checklists used by the trained farmer. All diseases were graded as mild, moderate, or severe diseases based on the scoring system described elsewhere [30]. For statistical analysis, the disease scoring was dichotomized from three categories (mild, moderate, severe) to two categories (moderate, severe), whereby “mild disease” was changed to “moderate”. Moderate and severe deviations were summarized as “severe”.

#### 2.7.1. Calf Diarrhea

Following the SOPs, defecation was assessed by the farmers for type, frequency, and painfulness. Fecal consistency, color, and odor, the degree of digestion, and foreign matter were evaluated. The farmers noted the fecal consistency. The consistency may be categorized as described elsewhere: firm, soft, mushy, liquid, and watery [30] (pp. 147–148). To simplify and unify the findings, the farmers documented the feces as “firm” (=moderate disease), “mushy” (=moderate disease), or “watery” (=severe disease).

#### 2.7.2. Navel Illness

The umbilicus was assessed extra- and intra-abdominally by palpation. During the physical examination on the 3rd to the 6th days of age by one of the authors (C.H.), any sign of inflammation (swelling/changed consistency, heat, pain) and any signs of an umbilical or inguinal hernia or enlarged umbilical arteries/veins were noted [31]. To create a suitable SOP for farmers, only navel thickness (assessed by palpation) was assessed during the shortened daily physical examinations. As described in the clinical propaedeutic, navel thickness was assessed using eight categories [30,31]. For the statistical analysis, the pathological findings were classified as moderate or severe navel illness, defined as two-finger-thick and three-or-more-finger-thick swellings, respectively.

#### 2.7.3. Bovine Respiratory Disease

As described in the SOPs, farmers assessed respiration by counting respiratory rates, coughing, and/or nasal discharge. Physiological respiration was defined as 20 to 40 calm and regular breaths per minute. Pathological changes were classified as follows: fast and shallow breathing (40 to 50 times per minute) was defined as moderate disease, very fast breathing, almost panting, and/or respiratory distress was defined as severe disease [30] (pp. 118–121). Coughing was divided into sporadic coughs (moderate disease) and intermittent coughing, which means more than 3 consecutive coughs (severe disease) [30] (p. 113). The quality of nasal discharge was assessed and categorized as serous discharge (moderate disease) if it had a watery, clear, slightly yellowish and/or greyish appearance. Severe disease was defined as viscous mucous that looked opaque and/or purulent discharge showing a yellow color [30] (p. 90).

#### 2.7.4. Abnormal Behavior

Normal physiological behavior was defined as a bright and alert appearance [30] (p. 54). Pathological findings were classified according to three categories defined as follows: mildly depressed/somnolent (moderate disease), moderately depressed/stupor (severe disease), and severely depressed/coma (severe disease). As per these definitions, mildly depressed calves appear dull, apathetic, and sleepy. When walking, slight swaying is observed, and the eyelids are partially closed. Feed and water intake is reduced. Moderately depressed calves lie down determinedly and show markedly sleepy behavior. They can only be roused by very strong stimuli. Calves with severe depression have delayed reflexes and are recumbent, and their respiration is superficial [30] (p. 55).

### 2.8. Statistical Analysis

Data were collected and summarized using Microsoft Excel 2016. Subsequently, the data were transferred to IBM^®^ SPSS^®^ Statistics Version 28 (IBM^®^, New York, NY, USA) for further statistical analysis. Descriptive statistics were carried out and values are shown as the median 25th and 75th percentiles, minimum and maximum, since the Brix values were not normally distributed (*p* < 0.05, Kolmogorov–Smirnov test including Lilliefors correction). The coded values were labeled and factors that might explain an insufficient immunoglobulin concentration in the colostrum and in the calf serum were investigated using a two-step process. The first step was a binary logistic regression and the second step was a multiple logistic regression searching for statistically significant associations (*p* < 0.05). The binary logistic regression was carried out using the dichotomous colostral Brix values (>22%; ≤22%) and serum Brix values (<8.4%; ≥8.4%) as dependent variables. The statistical procedure has been described elsewhere [4]. All factors that might influence the outcome were included as covariates. The binary and multiple logistic regression analyses showed odds ratios (OR) with a 95% confidence interval (95% CI) for the different categories of the covariates compared to the defined reference category. If the OR showed results equal to 1 (OR = 1), there was no association between the factor and the outcome. Factors with odds ratios > 1 (OR > 1) were interpreted as factors that were associated with the outcomes (colostrum quality: >22% or ≤22% Brix; serum brix values: <8.4%; ≥8.4%). Factors with odds ratios < 1 (OR < 1) were interpreted as protective factors. The associations between a low serum Brix value, which was defined as FTPI (Brix < 8.4%), and the occurrence of calf diarrhea, BRD, navel illness, and/or an abnormal general behavior were investigated using disease occurrence (healthy, moderate disease, severe disease) as covariates. Case definitions were used as described in Section 2.7. If the OR showed results equal 1 (OR = 1), there was no association between disease occurrence and the outcome (serum Brix values: <8.4%; ≥8.4%). Factors with odds ratios > 1 (OR > 1) were interpreted as factors that were associated with variable outcomes. Factors with odds ratios < 1 (OR < 1) were interpreted as protective factors. The Hosmer–Lemeshow goodness-of-fit function was applied for all analyses. All implemented tests were interpreted as statistically significant if the *p* value was <0.05.

## 3. Results

### 3.1. Colostrum Quality

Of the investigated colostrum samples, 140 (57.1%) showed poor colostrum quality of ≤22% and 105 (42.9%) a good colostrum quality of >22% (Figure 1). The median colostrum Brix values were 21.2% (min = 7.3%, max = 35.1%, 25th = 18.5%, 75th = 25.0%). Primiparous cows (N = 71) showed median Brix values of 20.9% (min = 7.3%, max = 32.2%, 25th = 16.4%, 75th = 24.2%). Cows in their second (N = 58) and third (N = 49) lactations showed median Brix values of 20.9% (min = 10.9%, max = 28.3%, 25th = 18.0%, 75th = 24.0%) and 21.1% (min = 13.0%, max = 35.1%, 25th = 18.8%, 75th = 23.5%), respectively. All cows with >3 lactations (N = 66) were summarized and processed as one group and yielded median Brix values of 23.8% (min = 11.9%, max = 31.0%, 25th = 20.5%, 75th = 26.2%).

#### 3.1.1. Explanatory Variables for Colostrum Quality

##### Binary Logistic Regression

Overall, the median number of lactations was 2.0 (min = 1.0, max = 11.0, 25th percentile = 1.0, 75th percentile = 4.0). The median gestation length (N = 234) was 284 days (min = 262, max = 301, 25th = 280, 75th = 288). The median time lag between parturition and the first milking (N = 248) was 75 min (min = 0, max = 960, 25th = 30.0, 75th = 180.0). The median dry period length, excluding primiparous cows (N = 174, missing information = 1), was 8.0 weeks (min = 5, max = 23, 25th = 7.0, 75th = 12.0). The univariable approach showed that the number of lactations, the dry-off procedure, the dry period length, the time to first milking, and ante-partum milk leakage were significantly associated with colostrum quality. The association between coliform counts and colostrum quality was not evaluated, since all of the 135 investigated colostrum samples were below the threshold of ≤10,000 cfu/mL. Of these, 83 (61.5%) and 52 (38.5%) showed Brix values ≤ 22% and >22%, respectively. The results of the binary logistic regression for colostrum quality are summarized in Table 1.

##### Multiple Logistic Regression Colostrum Quality

After the exclusion of the primiparous cows (no information on the dry period length and no dry-off procedure available) and missing values, information on 173 multiparous cows was available. The number of lactations (>3 lactations) turned out to be significantly associated with good colostrum quality (OR = 0.18, 95% CI = 0.04–0.73, *p* = 0.02). The time lag between parturition and the first milking was statistically associated with poor colostrum quality; for details, see Table 2.

### 3.2. Physical Examination of the Calves

Milk intake was very good, good, moderate, poor, or absent in 215 (86.0%), 21 (8.4%), 8 (3.2%), 4 (1.6%), and 2 (0.8%) calves, respectively. The median body temperature was 38.9 (min = 38.0 °C, max = 40.5 °C, 25th percentile = 38.7 °C, 75th percentile = 39.1 °C) and the median pulse rate was 117 beats (min = 80, max = 200, 25th = 111, 75th = 124). The detailed results of the physical examinations conducted from the third to the sixth day of age are provided in the Appendix A.

### 3.3. Serum and Plasma Brix Values

Of the 250 included calves, 93 (37.2%) and 157 (62.8%) showed an insufficient colostrum supply (FTPI) and a sufficient colostrum supply, respectively (Figure 2). The median serum and plasma Brix values were 8.6% and 9.3%, respectively. The serum and plasma Brix values ranged from 6.3% to 11.8% (25th percentile = 8.1%, 75th percentile = 9.3%) and 7.0% to 12.4% (25th = 8.8%, 75th = 10.1%), respectively.

#### 3.3.1. Explanatory Variables for Serum Brix Levels

##### Binary Logistic Regression

The median time lag between parturition and the first feeding (N = 244) was 100 min, (min = 10, max = 1260, 25th percentile = 45, 75th percentile = 195). The median time lag between the first milking and calf feeding was 15 min (min = 0, max = 660.0, 25th = 10.0, 75th = 30.0). The median quantity of colostrum intake was 2.5 L (min = 0, max = 5.0, 25th = 2.0, 75th = 3.0). The colostrum intake of the bottle-fed calves was categorized by the farmers as good in 221 calves and poor in 15 calves. Additionally, 3 calves were fed using an esophageal feeder and 11 calves stayed with their dam. In total, 234 calves received colostrum from their mother and 15 calves from other sources (a cow other than the mother, frozen colostrum stock), (missing value N = 1). The median TPC was 25,850 cfu/mL (min = 0, max = 3,030,000, 25th = 7000, 75th = 121,500). The association between coliform counts and serum Brix levels was not evaluable, since all of the 136 investigated colostrum samples were within the threshold with ≤10,000 cfu/mL. Of these, 57 (41.9%) and 79 (58.1%) showed Brix values < 8.4% and ≥ 8.4%, respectively. The results are summarized in Table 3. The median coliform counts were 0 (min = 0, max = 2150, 25th = 0, 75th = 10).

##### Multiple Logistic Regression

After applying multiple logistic regression, the colostrum quality, time to first feeding, and amount of colostrum fed to the calf showed a significant association with low serum Brix levels, indicating FTPI. For details, see Table 4.

### 3.4. Effect of FTPI on Neonatal Diseases

Of the 250 included calves, 45 calves (18.0%), 44 calves (17.6%), and 25 calves (10.0%) experienced a health event of any kind in the first, second, or third week of life, respectively. Initially, health events (any signs of neonatal disease, as defined in Section 2.7) were summarized on a weekly basis and the associations between disease and FTPI were analyzed (for details, see Table 5). The association between disease occurrence and FTPI was the strongest in the third week; for details, see Table 5. Regarding the investigated neonatal diseases, only diarrhea showed a statistically significant association with an odds ratio of 2.69 (95%CI = 1.52–4.76); for details, see Table 6. For BRD, navel illness, the combination of diarrhea and BRD, and altered general behavior, the occurrence was too low for any statistically significant differences to be determined. When calculating the odds ratios for each week (1st, 2nd, and 3rd) for each disease, only diarrhea showed an association with FPTI. In the first week of life, the OR was 2.26 (95%CI = 1.10–4.65), in the second week it was 2.84 (95%CI = 1.38–5.85), and in the third week it was 4.61 (95%CI = 1.81–11.76), including the healthy calves as the basic category (OR = 1). The onset of severe diarrhea in the second and third weeks of life was particularly strongly associated with FTPI with odds ratios of 6.1 (1.6–23.26) and 16.13 (1.98–125.0), respectively. Detailed information on the associations between the occurrence of neonatal diseases and low serum Brix levels are provided in the Appendix A. In the multiple logistic regression, only the occurrence of diarrhea in the first 3 weeks was statistically significantly associated with a low serum Brix level of <8.4% (OR = 3.3, 95%CI = 1.77–6.13, *p* < 0.001).

## 4. Discussion

In total, 250 calves from 11 dairy farms from the federal state of Salzburg (Austria) were included in the study. Of the 245 investigated colostrum samples, 140 (57.1%) showed were of poor quality, with ≤22% Brix. This is similar to the findings from other countries, where poor-quality colostrum has been found in 15.5% to 57.8% [4,14,29] of samples. The aforementioned studies investigated the colostrum samples using radial immunodiffusion (RID). Radial immunodiffusion is currently recognized as the gold standard; elsewhere, it has been noted that cheap and user-friendly indirect measurement methods, such as the digital Brix refractometer, are reliable alternatives for evaluating colostrum and serum immunoglobulin concentrations [15,32,33]. Of the 250 included calves, 93 (37.2%) showed a low serum Brix level of less than 8.4%, indicating FTPI. In plasma, the threshold for FTPI remains higher. An investigation carried out in Germany of the different analytical approaches for assessing FTPI described thresholds of 7.8% for serum and of 8.6% for plasma [15]. The occurrence of FTPI found in this study is comparable to that identified in other investigations, where FTPI was found in between 14.1% and 41.9% of the included colostrum samples [4,14,15,17,18,19]. There are differences between countries and herds regarding the occurrence of low-quality colostrum and FTPI [19]. In the present study, the best farm had 88.5% high-quality colostrum samples (Farm E) and the worst farm had 13.3% (Farm G) high-quality colostrum samples. Similarly, the number of calves showing a sufficient colostrum supply was 84.6% for the best farm (Farm E) and 26.7% for the worst farm (Farm G). It has been shown that individual farm management factors have a significant impact on FTPI frequency. A study conducted in Quebec (Canada) investigated the herd-level prevalence of FTPI in 59 dairy herds (with a minimum of 14 calves included per herd) and described high herd-level variations in the rates of calves experiencing FTPI, ranging between 30% and 100% [34]. A study conducted in Italy included 21 farms (244 calves) and described a within-herd prevalence of FTPI of 20.0% to 71.4% [35]. This underlines the importance of herd-specific evaluations of colostrum management practices.

In the present study, the variation in colostrum quality was high, with minimum levels of 7.3% and maximum levels of 35.1%. Since good colostrum quality acts as a protective factor for FTPI (OR = 0.26), it is essential to test the colostrum’s quality before delivering it to the calf [4]. In an online questionnaire on calf management practices in Austria, only 20.8% of the 1287 included farmers implemented colostrum testing protocols on their farms; of these farmers, 86.1% stated that they perform visual inspection of the colostrum [36]. There is a further need to increase farmers awareness of good colostrum management practices, in which colostrum testing plays an essential role. Another management-related factor proven to be significantly associated with colostrum quality is the time lag between parturition and the colostrum harvest. Harvesting colostrum 2 to 6 h after parturition showed significantly negative effects on colostrum quality in comparison with colostrum harvested within the first 2 h after parturition (OR = 5.29). Colostrum IgG levels have been described to decrease by 3.7% every hour after parturition [20]. Therefore, it is essential to milk the colostrum as soon as possible after birth. The positive effect of cows having more lactations on colostrum quality has been proven in multiple investigations on colostrum management [29]. In the present study, cows in their second lactation showed the worst colostrum quality, which is in accordance with other studies [4,29,37]. The binary logistic regression showed that ITS acts as a protective factor for poor colostrum quality (OR = 0.21). However, in the present study, this factor was not significant in the multiple logistic regression. It can be hypothesized that this might be due to the reduced probability of milk leakage ante partum and/or during parturition. In the present study, the questionnaire solely considered whether the cow experienced milk leakage during parturition and/or immediately before the colostrum harvest. Further studies are needed to investigate whether ITS might improve colostrum quality. It has been proven elsewhere that bacterial contamination of the colostrum has an effect on the apparent efficiency of immunoglobulin absorption and on the probability of calves experiencing pneumonia [5,38]. Only a few of the investigated colostrum samples showed contamination with coliform bacteria; therefore, it was not possible to calculate the effects on colostrum quality. Additionally, the majority of the TPC values were below the published threshold of <100,000 cfu/mL and the comparative groups were too small [2]. Due to resource issues, it was not feasible to investigate 100% of the colostrum samples for bacterial contamination; further work is needed in this field.

Factors such as feeding the calf with high-quality colostrum (>22% Brix) and higher quantities of colostrum (≥2 L) were revealed to be protective factors for calves facing FTPI, yielding odds ratios of OR = 0.16 and OR = 0.25, respectively. These effects have been proven several times by different research groups and are among the most important factors for ensuring a successful colostrum supply [1,39]. Contradictory results were found regarding the time to first feeding where the time lag (>360 min) seem to be protective (OR = 0.21). In total, 51.2% of the calves were fed within the first 2 hours of life, only 37.7% between 2 and 6 h after parturition, and only 11.1% more than 6 h after parturition. The number of calves in the last group was small and the distribution between calves with and without FTPI was unequal. Therefore, this result must not be over-interpreted. The common recommendations for good colostrum management state that the calf should be fed immediately after parturition [1].

In total, the number of calves showing any signs of BRD, navel illness, diarrhea and BRD, and/or an abnormal behavior was low. Therefore, it was not feasible to assess the association between each disease in the first, second, or third week of life. When summarizing any neonatal disease as a “health event”, there was a clear association (OR > 1) between disease occurrence and FTPI. The consequences of FTPI in terms of adjusted risks for diarrhea and BRD were found to be 1.75 and 1.51 in a recent meta-analysis [10]. With the present investigation, we can support the finding that there is a clear association between diarrhea and FTPI, with an odds ratio of 2.69 (1.52–4.76). The findings underline the importance of protecting calves from FTPI to prevent them from contracting neonatal diseases, especially diarrhea. In addition to successful colostrum management, there are multiple management-related, calf-related and environment-related factors that need to be addressed to achieve low on-farm calf morbidity and mortality rates.

In the present study, the farmers showed high levels of commitment and carefully followed the SOPs for physical examinations after receiving detailed training from one of the first authors (C.H.). One of the major limitations of the study was that no sample size calculation could be carried out in advance, since the farms joined the study voluntarily. Therefore, we cannot exclude an over- or under-representation of FTPI. Additionally, the number of calves suffering from pneumonia and navel illness was too low for us to assess the impact of FTPI. Further studies are needed to obtain information on the true prevalence of FTPI in Austria and to further assess the impact of FTPI on neonatal diseases. Brix refractometry was used as a reference method to indirectly test for immunoglobulin concentrations in colostrum and serum. In future investigations, we recommend implementing the RID, which is currently recognized as the gold standard. In consensus with the literature from Austria, there is potential to improve colostrum management on farms in Austria, which could be addressed by the Animal Health Service. Good colostrum management is an indispensable component of raising healthy youngstock. In Austria, vets and advisers should not just focus on the effective transfer of passive immunity in calves but rather focus on achieving excellent passive transfer in calves and further reducing morbidity and mortality [9].

## 5. Conclusions

Colostrum harvested from cows within 2 hours of parturition and the colostrum from cows with >3 lactations was of a significantly better quality (>22% Brix). In the present study, the colostrum quality (>22% Brix) and the colostrum quantity (≥2 L) were the most important factors in preventing calves from experiencing FTPI. There is a strong association between the occurrence of calf diarrhea and FTPI within the first three weeks of life. Therefore, a sufficient colostrum supply achieving serum Brix levels greater than 8.4% is essential to protect calves from disease.

## Figures and Tables

**Figure 1 animals-13-01740-f001:**
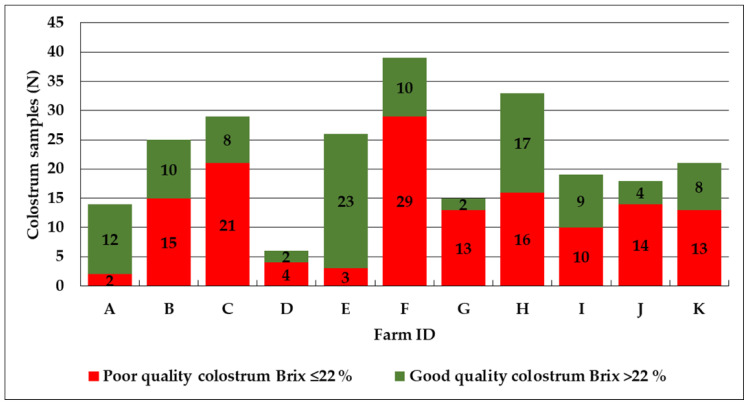
Overview of the 245 colostrum samples originating from 11 dairy farms of the Austrian federal state of Salzburg. The colostrum Brix values were categorized as high-quality colostrum (>22%, green color) and low-quality colostrum (≤22%, red color).

**Figure 2 animals-13-01740-f002:**
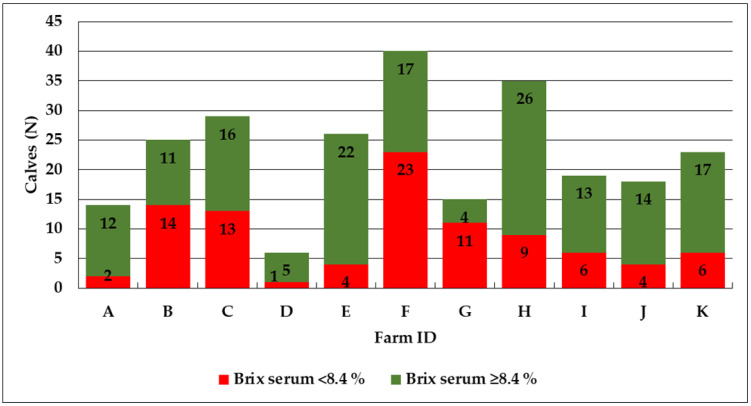
Overview of 250 serum samples from calves originating from 11 dairy farms of the Austrian federal state of Salzburg. The serum Brix values were categorized as a sufficient colostrum supply (≥8.4%, no FTPI, green color) and an insufficient colostrum supply (<8.4%, FTPI, red color), respectively.

**Table 1 animals-13-01740-t001:** Binary logistic regression for factors associated with poor colostrum quality. Poor colostrum quality was defined as a Brix value of ≤22%. Odds ratios (OR) were calculated using the basis category (OR = 1) as a reference. Odds ratios < 1 were interpreted as protective factors for poor colostrum quality and odds rations > 1 were interpreted as factors associated with poor colostrum quality. All statistically significant differences (*p* < 0.05) are highlighted with an asterix * (AB, antibiotic intramammary treatment; ITS, internal teat sealant).

Factor	N Samples Total	Brix ≤ 22%		Brix > 22%		OR (95% CI)
		N Samples	% Samples	N Samples	% Samples	
Number of lactations						
1	71	48	67.6	23	32.4	1
2	58	38	65.5	20	34.5	0.91 (0.44–1.90)
3	49	32	65.3	17	34.7	0.91 (0.42–1.94)
>3	66	22	33.3	44	66.7	0.24 (0.12–0.49) *
Missing	6					
Gestation length						
0–279 days	37	19	51.4	18	48.6	1
>280 days	192	111	57.8	81	42.2	1.30 (0.64–2.63)
Missing	21					
Ante-partum milk leakage						
Yes	53	37	69.8	16	30.2	1
No	191	102	53.4	89	46.6	0.50 (0.26–0.95) *
Missing	6					
Diseases during dry period						
Yes	11	6	54.5	5	45.5	1
No	234	134	57.3	100	42.7	1.12 (0.33–3.76)
Missing	5					
Time to first milking						
0–119 min	141	65	46.1	76	53.9	1
120–359 min	72	52	72.2	20	27.8	3.04 (1.65–5.61) *
≥360 min	32	23	71.9	9	28.1	2.99 (1.29–6.91) *
Missing	5					
Colostrum harvested						
0–3 L	109	59	54.1	50	45.9	1
4–6 L	104	64	61.5	40	38.5	1.36 (0.79–2.34)
>6 L	29	15	51.7	14	48.3	0.91 (0.40–2.06)
Missing	8					
Total plate count						
TPC < 100,000/mL	100	63	63.0	37	37.0	1
TPC ≥ 100,000/mL	38	23	60.5	15	39.5	0.90 (0.42–1.94)
Missing	112					
Dam vaccination						
Yes	30	22	73.3	8	26.7	1
No	215	118	54.9	97	45.1	0.44 (0.19–1.04)
Missing	5					
Dry-off procedure						
Primiparous cow	69	46	66.7	23	33.3	1
AB	92	60	65.2	32	34.8	0.94 (0.48–1.81)
ITS	37	11	29.7	26	70.3	0.21 (0.09–0.50) *
No medication	11	5	45.5	6	54.5	0.42 (0.11–1.51)
AB+ITS	34	17	50.0	17	50.0	0.5 (0.22–1.16)
Missing	7					
Dry period length						
Primiparous cow	70	48	68.6	22	31.4	1
<8 weeks	59	28	47.5	31	52.5	0.41 (0.20–0.85) *
≥8 weeks	115	64	55.7	51	44.3	0.58 (0.31–1.07)
Missing	6					

**Table 2 animals-13-01740-t002:** Multiple logistic regression for factors associated with poor colostrum quality. Poor colostrum quality was defined as a Brix value of ≤22%. Odds ratios (OR) were calculated using the basis category (OR = 1) as a reference. Odds ratios < 1 were interpreted as protective factors for poor colostrum quality and OR values > 1 were interpreted as factors associated with poor colostrum quality. All statistically significant differences (*p* < 0.05) are highlighted with an asterix * (AB, antibiotic intramammary treatment; ITS, internal teat sealant).

	OR (95% CI)	*p* Value
Number of lactations		
1	1	
2	0.68 (0.17–2.73)	0.59
3	0.71 (0.18–2.85)	0.62
>3	0.17 (0.04–0.70)	0.01 *
Gestation length		
0–279 days	1	
>280 days	1.36 (0.28–6.45)	0.70
Ante-partum milk leakage		
Yes	1	
No	0.70 (0.24–1.99)	0.50
Diseases during the dry period		
Yes	1	
No	0.35 (0.05–2.26)	0.27
Time to first milking		
0–119 min	1	
120–359 min	5.35 (1.83–15.63)	0.00 *
≥360 min	5.43 (1.48–20.00)	0.01 *
Colostrum harvested		
0–3 L	1	
4–6 L	1.18 (0.46–3.04)	0.74
>6 L	0.37 (0.09–1.52)	0.17
Total plate count		
TPC < 100,000/mL	1	
TPC ≥ 100,000/mL	0.57 (0.21–1.50)	0.25
Dam vaccination		
Yes	1	
No	0.58 (0.17–1.98)	0.39

**Table 3 animals-13-01740-t003:** Binary logistic regression for factors associated with low serum Brix levels (< 8.4%) indicating FTPI. Odds ratios (OR) were calculated using the basis category (OR = 1) as a reference. All statistically significant differences (*p* < 0.05) are highlighted with an asterix *. Assisted calving was defined as one person assisting at calving and severe dystocia was defined as more than one person assisting at calving and/or the vet being consulted.

Factor	N Calves Total	Brix < 8.4%		Brix ≥ 8.4%		OR (95% CI)
		N Calves	% Calves	N Calves	% Calves	
Colostrum quality						
≤22% Brix	140	69	49.3	71	50.7	1
>22% Brix	105	21	20.0	84	80.0	0.26 (0.14–0.46) *
Missing	5					
Total plate count						
<100,000 cfu/mL	101	44	43.6	57	56.4	1
≥100,000 cfu/mL	38	15	39.5	23	60.5	0.84 (0.40–1.81)
Missing	111					
Time to first feeding						
0–119 min	125	45	36.0	80	64.0	1
120–359 min	92	39	42.4	53	57.6	1.3 (0.75–2.27)
≥360 min	27	6	22.2	21	77.8	0.51 (0.19–1.35)
Missing	6					
Colostrum intake						
<2 L	43	23	53.5	20	46.5	1
≥2 L	190	61	32.1	129	67.9	0.41 (0.21–0.81) *
Missing	17					
Calving						
Unassisted	192	64	33.3	128	66.7	1
Assistance	40	18	45.0	22	55.0	1.64 (0.82–3.27)
Severe dystocia	8	5	62.5	3	37.5	3.33 (0.77–14.39)
Missing	10					

**Table 4 animals-13-01740-t004:** Multiple logistic regression for factors associated with low serum Brix values (<8.4%) indicating FTPI. Odds ratios (OR) were calculated using the basis category (OR = 1) as a reference. All statistically significant differences (*p* < 0.05) are highlighted with an asterix *. Assisted calving was defined as one person assisting at calving and severe dystocia was defined as more than one person assisting at calving and/or the vet being consulted.

Factor	OR (95% CI)	*p* Value
Colostrum quality		
≤22% Brix	1	
>22% Brix	0.16 (0.06–0.43)	<0.001 *
Total plate count		
<100,000 cfu/mL	1	
≥100,000 cfu/mL	0.81 (0.33–1.96)	0.64
Time to first feeding		
0–119 min	1	
120–359 min	0.83 (0.33–2.09)	0.69
≥360 min	0.21 (0.06–0.8)	0.02 *
Colostrum intake		
<2 L	1	
≥2 L	0.25 (0.08–0.76)	0.01 *
Calving		
Unassisted	1	
Assistance	2.09 (0.79–5.56)	0.14
Severe dystocia	3.77 (0.62–23.26)	0.15

**Table 5 animals-13-01740-t005:** In total, 250 calves were included in the study and any sign of disease, as defined in Section 2.7, was documented for the first three weeks once per day. Signs of diarrhea, navel illness, BRD, and abnormal general behavior, as well as disease combinations, were summarized as “health events” in the binary logistic regression. The odds ratios (OR) were calculated for associations with FTPI (Brix values < 8.4%). Odds ratios were calculated using the basic category = healthy calves (OR = 1) as a reference. All statistically significant differences (*p* < 0.05) are highlighted with an asterix *.

Factor	N Calves Total	Brix < 8.4%		Brix ≥ 8.4%		OR (95% CI)
		N Calves	% Calves	N Calves	% Calves	
First week						
Healthy	205	70	34.1	135	65.9	1
Health event	45	23	51.1	22	48.9	2.02 (1.05–3.87) *
Second week						
Healthy	206	70	34.0	136	66.0	1
Health event	44	23	52.3	21	47.7	2.13 (1.10–4.11) *
Third week						
Healthy	225	76	33.8	149	66.2	1
Health event	25	17	68.0	8	32.0	4.17 (1.72–10.10) *
First to third weeks						
Healthy	168	50	29.8	118	70.2	1
Health event	82	43	52.4	39	47.6	2.60 (1.51–4.49) *

**Table 6 animals-13-01740-t006:** Binary logistic regression analysis was carried out using the serum Brix levels as a dependant variable. The table gives an overview of the associations of diarrhea, bovine respiratory disease (BRD), navel illness, diarrhea and BRD, and abnormal general behavior with a low serum Brix level of <8.4% in the first 3 weeks of life. Odds ratios (OR) were calculated using the basic category = healthy calves (OR = 1) as a reference. All statistically significant differences (*p* < 0.05) are highlighted with an asterix *.

Factor	N Calves Total	Brix < 8.4%		Brix ≥ 8.4%		OR (95% CI)
		N Calves	% Calves	N Calves	% Calves	
Diarrhea						
Healthy	182	56	30.8	126	69.2	1
Diarrhea	68	37	54.4	31	45.6	2.69 (1.52–4.76) *
BRD						
Healthy	227	86	37.9	141	62.1	1
BRD	23	7	30.4	16	69.6	0.72 (0.28–1.81)
Navel illness						
Healthy	248	92	37.1	156	62.9	1
Navel illness	2	1	50.0	1	50.0	1.69 (0.10–27.78)
Diarrhea and BRD						
Healthy	240	91	37.9	149	62.1	1
Diarrhea + BRD	10	2	20.0	8	20.0	0.41 (0.09–1.97)
Abnormal general behavior						
Healthy	226	81	35.8	145	64.2	1
Abnorm. behavior	24	12	50.0	12	50.0	1.79 (0.77–4.17)

## Data Availability

Data are available within the article or in its Appendix A.

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
