# Peer review of "Factors Associated with Colostrum Quality, the Failure of Transfer of Passive Immunity, and the Impact on Calf Health in the First Three Weeks of Life"

_animals, 2023, doi:10.3390/ani13111740_

Round 1
Reviewer 1 Report
The aim study evaluate factors associated with colostrum quality and failure of transfer of passive immunity in calves from dairy farms in Austria and to assess associations between disease occurrence and failure of transfer of passive immunity in calves. Although this is only study run in one part of Austria, the results are important, as can be used to compare the situation in different countries.
My specific comments are:
-lines 36-37 “Calves facing any health event (diarrhea, navel illness, bovine respiratory disease, abnormal behavior) in the first three weeks of age had a higher probability of FTPI”
How probability was here calculated? Do not see in the text. Why you calculate probability of FTPI in sick calves? Wouldn't it make more sense to calculate the probability of disease in calves with FTPI, especially this would be in the line with your hypothesis no 3.
-lines 247-249 “Pathological findings were classified in three categories defined as follows: mildly depressed/somnolent (moderate disease), moderately depressed/stupor (severe disease) and severely depressed/coma (severe disease)”.
Why there are three categories listed but two of them are named –severe disease?
-lines 260-261 If there were not normally distributed than p of the K-S test was <0.05.
p-value < 0.05 means that the null hypothesis is rejected and the distribution is not normal.
Please check what was the p value in this case.
-Table 2 line numbers are on table 2 which makes difficult to read.
-line 366 “Colostrum intake was very good, good, moderate, poor or absent in...”
Define what where the categories for intake. I see only categories ≥2L or <2L
-lines 400-401 “The colostrum intake of the bottle-fed calves was categorized by 400 the farmers as good in 221 calves and poor in 15 calves” based on which criteria intake was evaluated?
The table in supplementary materials are in German, so if this criteria are defined there it should be translated into English.
-Table 3 results in Factor-TPC are not in the right position, move it one line below
-line 431 there are two table no 4 in this paper. Please correct table numbering
-table 4 from line 449 Heading If the OR is calculated for association of FPTI it should be written in the title of the table. Giving at the same moment results for Brix≥8.4 is confusing.
-table 5 Heading ‘low serum Brix level” should be specified (this is <8,4%)
-Discussion section- there is an excellent publication (see below) about protection level in calves (also in brix units) and morbidity. Please consider USA recommendation in the discussion section. Would it be possible to rich excellent protection in Austria conditions?
Paper: Lombard J, Urie N, Garry F, Godden S, Quigley J, Earleywine T, McGuirk S, Moore D, Branan M, Chamorro M, Smith G, Shivley C, Catherman D, Haines D, Heinrichs AJ, James R, Maas J, Sterner K. Consensus recommendations on calf- and herd-level passive immunity in dairy calves in the United States. J Dairy Sci. 2020 Aug;103(8):7611-7624. doi: 10.3168/jds.2019-17955. Epub 2020 May 21. PMID: 32448583.
Author Response
Dear Reviewer,
Please find our response in the attached word document.
Yours sincerely,
Katharina Lichtmannsperger (on behalf of all authors)

Reviewer 2 Report
Lines 34-35 Does this refer to colostrum produced by the cow and/or consumed by the calf? This should be more clearly distinguished.
Line 51 What is the nutrient advantage of colostrum versus milk?
Lines 288-289 Why is there only 245 colostrum samples but line 29 reports 250?
Lines 291-297 Primiparous cows' data is reported as range and then (median, 25th, 75th) while the rest are median then (min, max, 25th, 75th). Report this consistently between each group.
Line 310 Why were so few of the samples (135) tested for TPC and coliform counts?
Table 1 Gestation length 0-279 days has the wrong percent (37) for % of samples under Brix > 22%. Should be 48.6% (18/37).
Line 520 Which category of "time to first feeding" did the calves that remained with their dam fall under? Could this have played a role in the group that were > 360 minutes. Did you compare these calves to those that did not remain with their dams?
Line 16-17 This sentence could be two sentences with the second saying, "Therefore, FTPI poses a major animal welfare issue."
Line 487 should start as follows, "In the present study,..."
Line 526 Start a new paragraph with, "In total, ..."
Author Response

(The authors gave the same response as above.)
